# From Patient to Musician: A Multi-Sensory Virtual Reality Rehabilitation Tool for Spatial Neglect

Joris Heyse [1,*,†], Stéphanie Carlier [1,*,†], Ewoud Verhelst [1], Catharine Vander Linden [2], Femke De Backere [1] and Filip De Turck [1]

1   IDLab, Department of Information Technology, Ghent University—imec, Technologiepark-Zwijnaarde 126, 9052 Ghent, Belgium; ewoud.verhelst@ugent.be (E.V.); femke.debackere@ugent.be (F.D.B.); filip.deturck@ugent.be (F.D.T.)
2   Department of Rehabilitation Sciences, Ghent University Hospital, Corneel Heymanslaan 10, 9000 Ghent, Belgium; catharine.vanderlinden@ugent.be
*   Correspondence: joris.heyse@ugent.be (J.H.); stephanie.carlier@ugent.be (S.C.)
†   These authors contributed equally to this work.

**Abstract:** Unilateral Spatial Neglect (USN) commonly results from a stroke or acquired brain injury. USN affects multiple modalities and results in failure to respond to stimuli on the contralesional side of space. Although USN is a heterogeneous syndrome, present-day therapy methods often fail to consider multiple modalities. Musical Neglect Therapy (MNT) is a therapy method that succeeds in incorporating multiple modalities by asking patients to make music. This research aimed to exploit the immersive and modifiable aspect of VR to translate MNT to a VR therapy tool. The tool was evaluated in a 2-week pilot study with four clinical users. These results are compared to a control group of four non-clinical users. Results indicated that patients responded to triggers in their entire environment and performance results could be clearly differentiated between clinical and non-clinical users. Moreover, patients increasingly corrected their head direction towards their neglected side. Patients stated that the use of VR increased their enjoyment of the therapy. This study contributes to the current research on rehabilitation for USN by proposing the first system to apply MNT in a VR environment. The tool shows promise as an addition to currently used rehabilitation methods. However, results are limited to a small sample size and performance metrics. Future work will focus on validating these results with a larger sample over a longer period. Moreover, future efforts should explore personalisation and gamification to tailor to the heterogeneity of the condition.

**Keywords:** Unilateral Spatial Neglect; Musical Neglect Therapy; Virtual Reality

## 1. Introduction

Approximately 25 to 30% of all stroke individuals suffer from Unilateral Spatial Neglect (USN) [1]. Moreover, USN is a syndrome that is systematically under-diagnosed due to a lack of sensitivity in the used assessment methods [2]. Stroke is the leading cause of disability according to the World Health Organisation (WHO), and neglect predicts poor outcomes in functional recovery, leading to longer hospitalisations, functional dependency or long-term disability in everyday activities. Therefore, USN is an important syndrome that requires careful diagnosis and treatment [1,3].

USN is a syndrome that is commonly caused by unilateral brain injury, most often a stroke [4]. It is defined as the failure to report, respond or orient to meaningful or novel stimuli that occur on the side contralateral to the hemispheric lesion, where this failure cannot be attributed to either an elemental sensory or motor deficit [5]. The different modalities in which this lack of awareness manifests itself are manifold, and its presence severely hinders functional recovery after brain injury [6].

Moreover, USN is considered to be a heterogeneous syndrome, which means that the disorder can affect various modalities [7]. The most prevalent ones are the visual, auditory

and motor modality [8]. Although it is crucial for a rehabilitation method to include all the affected modalities, more often than not, solely the visual modality receives attention in current treatments [9]. Visual Scanning Training (VST), for example, is a technique that enjoys a high adoption rate in clinical practice but fails to incorporate other sensory modalities [10]. However, Music Neglect Therapy (MNT) is a therapy method based on active music-making that shows potential for the multi-sensory rehabilitation of USN [11].

The aim of this paper is to research the feasibility of translating MNT to a Virtual Reality (VR) environment for the rehabilitation of patients with USN, exploiting the immersive power and versatility of VR to provide a multi-sensory experience. For the design of this VR therapy tool, an interdisciplinary approach is taken in close cooperation with experts from the Ghent University Hospital. The resulting proof of concept is evaluated in a pilot study with four individuals suffering from USN and four non-neglect individuals.

The remainder of this paper is structured as follows. First, relevant related research is discussed in Section 2, followed by the vision of the therapy tool for USN in Section 3. Hereafter, the materials and methods will be discussed, subdivided into Section 4, which elaborates on the design of the proposed therapy tool, and Section 5, which discusses the implementation of the system. Next, Section 6 presents the setup of the performed pilot study. We follow with Sections 7 and 8, which present and discuss the findings of the pilot study. Finally, a concise conclusion will be given in Section 9.

## 2. Background

This section will discuss relevant background information and related work concerning the classification of spatial neglect, followed by the assessment and existing rehabilitation techniques for USN. Finally, musical therapy and the use of VR for patients with USN will be covered in depth.

### 2.1. Classification of Spatial Neglect

USN is relatively common and has many causes, including neurodegenerative diseases, neoplasm, aneurysm, traumatic brain injury after the surgical resection of a brain tumour and stroke, where stroke is by far the most prevalent cause [4,12–14]. Left-sided neglect is more common than right-sided neglect, as it is mostly linked to damage to the right hemisphere of the brain [15,16]. The brain has an asymmetric attention mechanism, where the left hemisphere is responsible for processing input originating from the right side of space, whereas the right hemisphere is responsible for stimuli from both the left and partly the right. When damage occurs at the left hemisphere, the right hemisphere can partially compensate for it. However, when the right hemisphere is damaged, no compensation is possible.

Although the existing literature is not consistent on the classification of the types of USN, two categories exist. The first categorises neglect based on the affected modality, including sensory, motor or representation neglect. Patients with sensory neglect experience a disordered awareness of sensory inputs originating from their contralesional side of space. They lack the ability to bring their attention to this side of space in the presence of external stimuli [12,15]. This type of neglect can be further specified by the sensory modality in which it manifests itself, such as visual, auditory and tactile neglect [8]. An example of sensory neglect is when the patient only eats half the plate of food, despite still being hungry [17]. Next, motor neglect is defined as a failure to generate the desired movement in response to observed stimuli, where this failure cannot be ascribed to a primary sensor or motor deficit [8,18]. Multiple sub-types of this type of neglect exist, such as the under-utilisation of a limb opposite to the brain lesion [19]. A final type of modality-based neglect is representation neglect or imagery neglect, where a person ignores the contralesional side of images that were internally generated, such as mental representations of a task, action or movement [8,20]. An example of this type of neglect can be seen when a patient is asked to draw an analogue clock from memory, in which the patient places all the numbers on their non-neglected side of the clock [21]. An example of this experiment is shown in Figure 1.

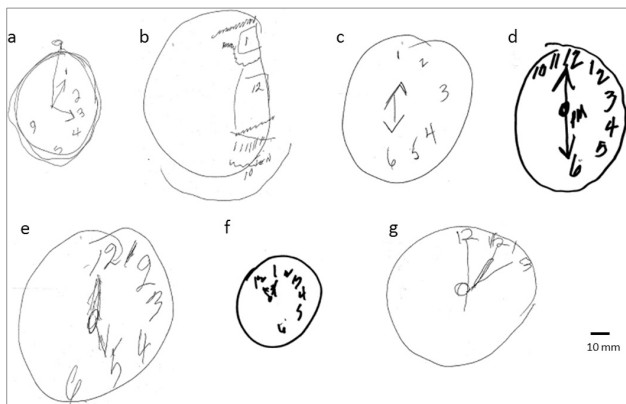

**Figure 1.** When patients with representational neglect are asked to draw an analogue clock from memory, the numbers are placed in their non-neglected side of the clock (**a**–**g**). Reprinted with permission from [21]. Copyright 2022 Journal of Neuropsychology.

The second method classifies neglect according to the distribution of abnormal behaviour in space, e.g., personal and spatial neglect [22]. Personal neglect is the lack of exploration of half of the body contralateral to the damaged hemisphere [23]. An example of personal neglect is only shaving half of their face, or putting on only one shoe. Personal neglect differs from sensory neglect as it is characterised by an unawareness of the body part itself, whereas sensory neglect refers to an unawareness of external stimuli, e.g., vision or touch [8]. Spatial neglect is defined as neglect where the abnormal behaviour is not located in personal space; this thus refers to failure to acknowledge stimuli on the contralesional side of space [23]. Spatial neglect is often used as a synonym for the neglect syndrome as a whole. Two sub-types of spatial neglect can be discerned. Peripersonal or near-space neglect is when symptoms manifest in the reaching distance of the patient, e.g., the example of eating half a plate of food. When symptoms occur in the space beyond reaching distance, e.g., inadvertently bumping into obstacles while walking, it is classified as extrapersonal or far-space neglect.

Although these categories exist, USN remains an inherently heterogeneous syndrome, meaning that patients often experience multiple types of neglect simultaneously [7].

*2.2. Assessment of USN*

To assess the degree and nature of the spatial neglect, several assessment methods are available today that try to measure a potential bias towards the non-neglected side of space. These assessment tests can be categorised into the so-called paper-and-pencil tests, which are often tools for quickly screening patients, and the behavioural tests, which aspire to assess spatial neglect based on everyday tasks [8,24,25].

Examples of paper-and-pencil tests are the line bisection test and the cancellation test. During the line bisection test, patients are asked to indicate the horizontal centre of a simple straight line (https://www.strokengine.ca/en/assessments/line-bisection-test/, accessed on 19 December 2021). A bias towards the right or left can then indicate USN, as shown in Figure 2 [26–28]. For the cancellation test, the patient has to cross out certain symbols on a sheet of paper, often ignoring other distraction symbols. One example of such a cancellation test is the star cancellation test, shown in Figure 3, in which all small stars need to be indicated, ignoring the random distraction words (https://strokengine.ca/en/assessments/star-cancellation-test/, accessed on 19 December 2021). The quantification of USN is then the difference between the missed stars on the left versus the right side [28–31]. An example of a digital paper-and-pencil test is the neglect test of the Test for Attention Performance (TAP). The neglect test is a subtest of the TAP test suite (https://www.psytest.net/en/test-batteries/tap/subtests#visual_field_examination, accessed on 19 December 2021), during which the patient needs to respond to a flickering stimulus on the screen by pressing a button [32]. These paper-and-pencil tests are simple and quick to set up.

However, they should only be used for initial screening and not for clinical diagnosis as they cannot discriminate between sensory and motor neglect.

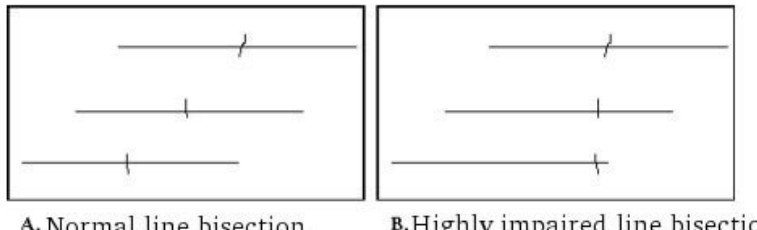

**Figure 2.** For the line bisection test, the middle of the line has to be indicated (**A**). A bias towards one of the extremities of the line can be an indication of USN (**B**).

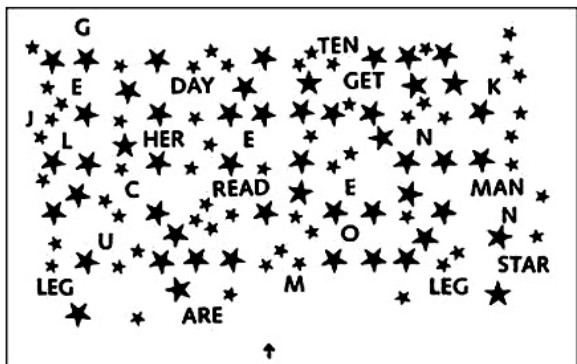

**Figure 3.** For the star cancellation test, all the stars have to be indicated. The difference in missed stars between the left and right side indicates potential USN.

Examples of behavioural tests are the Behavioural Inattention Test (BIT), the Semi-Structured Scale for Functional Evaluation of Hemi-Inattention and the Catherine Bergego Scale (CBS). The BIT consists of conventional pen-and-paper tests as well as practical tests such as reading a menu or sorting coins [8,28,33]. Similarly, for the Semi-Structured Scale for Functional Evaluation of Hemi-Inattention (https://strokengine.ca/en/assessments/semi-structured-scale-for-the-functional-evaluation-of-hemi-inattention/, accessed on 19 December 2021), the patient has to perform functional tasks such as combing their hair or serving tea [8,28,34]. Finally, the CBS is a standardised checklist for the detection and degree of unilateral neglect during the observation of everyday life situations (https://strokengine.ca/en/assessments/catherine-bergego-scale-cbs/, accessed on 19 December 2021). Moreover, the scale also measures the self-awareness of behavioural neglect. Examples are grooming and paying attention to people talking to the patient from one side of their field of view. The disadvantage of these behavioural tests is that they often focus on visually based functional tasks and do not differentiate between sensory and motor neglect [8,28].

### 2.3. Rehabilitation Techniques for USN

Many interventions for the rehabilitation of spatial neglect have been proposed; however, there is a need for more evidence regarding their effectiveness. Consequently, no standardised set of rehabilitation methods for USN exists [30]. The existing approaches can be categorised into top-down methods, bottom-up methods, modulation of inhibitory processes and increasing arousal.

Top-down methods aim to stimulate the voluntary attention gaze of the patient towards the neglected side of space. This stimulation consists of cues and instructions from the therapist. VST is the most used form of top-down rehabilitation technique. VST aims to orientate the visual scanning of the patient back towards the neglected side by giving

explicit instructions on where to look, e.g., asking them to locate the left-hand margin of the page before reading the next line [10]. Top-down methods are often criticised for their lack of specificity, as not all neglect symptoms seem to improve and non-visual sensory neglect is not addressed [35,36]. Nevertheless, top-down approaches, and specifically VST, are still mainly used in clinical practice [37].

In an attempt to address the shortcomings of top-down methods, bottom-up methods were developed. A deficit in processing the sensory information of one side of space creates an unbalance in the patient's representation of space. Bottom-up methods aim to restore this balance by manipulating the patient's sensory environment [25,38]. Sensory Stimulation is an example of such a method, which uses sensory input from different modalities, thereby addressing the multi-sensory aspect of USN, to eliminate this orientation bias towards the neglected side. The most well-known implementations of Sensory Stimulation are Optokinetic Stimulation (OKS), Neck-Muscle Vibration (NVM), Caloric Vestibular Stimulation (CVS) and Galvanic Vestibular Stimulation (GVS) [35]. Another well-known bottom-up technique is Prism Adaptation (PA), where patients wear prisms that shift their vision towards their non-neglected side. For left-sided neglect, the prism introduces an optical shift of several degrees to the right. At the start of the adaptation process, this leads to a right-sided error signal, as shown in Figure 4. After several iterations, compensation of the behaviour kicks in, and this error is corrected. When these prisms are then removed, a consistent deviation to the left occurs when pointing to a visual target, called the after-effect [39,40].

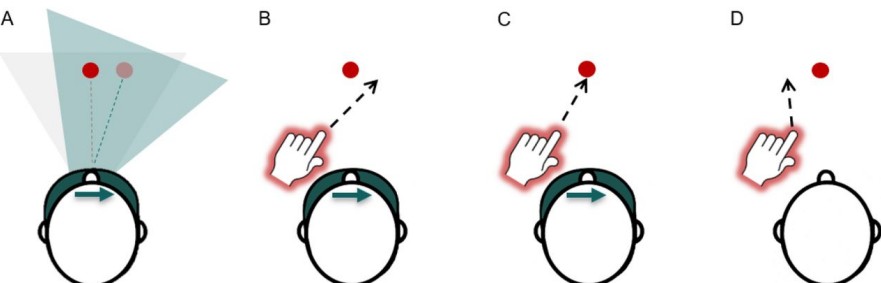

**Figure 4.** The four phases of the Prism Adaptation rehabilitation method. (**A**) Pre-test, (**B**) the perception of visual error signal, (**C**) the adaptation and (**D**) the after-effect. Reprinted with permission from [41]. Copyright 2022 Journal of Neuropsychology.

A final example of a bottom-up technique is Limb Activation (LA), where patients are asked to make voluntary contralesional limb movements in their affected side of space. This, in turn, activates the corresponding areas of the brain dealing with extra- and peripersonal space [42]. Studies regarding LA show interesting results; however, long-term functional effectiveness has yet to be determined [36].

The next category are methods that address the modulation of inhibitory processes, such as Transcranial Magnetic Stimulation (TMS) and transcranial Direct Current Stimulation (tDCS). These techniques use non-invasive brain stimulation to reduce the activity of the contralesional hemisphere or increase the activity of the ipsilesional hemisphere [43].

The final category of techniques for the rehabilitation of USN includes methods that focus on the stimulation of arousal, such as Sustained Attention Training (SAT) [7]. USN has been linked with a non-lateralised attention deficit, i.e., patients often struggle to keep their attention focused for a longer time period, even in their non-affected side of space [44]. These techniques aim to stimulate sustained attention to improve associated neglect symptoms. SAT does this by first pointing out the spatial errors that the patients makes during the executing of a certain task. The patient then has to repeat this task, while the trainer provides a reminder, every 20 to 40 s, to remain attentive, teaching the patient to "self-alert" while completing this task on their own [7].

### 2.4. Music Therapy and USN

Music therapy has shown great potential as a therapy method for multiple syndromes [45–47]. The act of playing an instrument provides a patient with the opportunity to train multiple sensory modalities simultaneously, such as hearing, vision, touch, motor and cognitive skills [48,49]. Moreover, music-making taps into the emotion and reward system of the human brain, making a music therapy intervention intrinsically rewarding and engaging [50].

Neurologic Music Therapy (NMT) is a neuroscientifically motivated model of music practice and consists of 20 research-based music therapy techniques. One of these techniques is MNT, a music-making spatial neglect therapy. In this model, MNT is described as a technique that "includes active performance exercises on musical instruments that are structured in time, tempo and rhythm, and is in appropriate spatial configurations, to focus the attention to a neglected or unattended visual field" [11].

The use of music in spatial neglect therapy has additional benefits, such as the Mozart effect. This effect states that one has better spatial–temporal reasoning when listening to a certain Mozart sonata [51], indicating that music and space are closely linked. It has been shown that, for people with spatial neglect, listening to classical music, or pleasant music in general, increases their spatial awareness and boosts arousal [52,53]. Second, people have a tendency to spatially represent ordinal sequences, such as numbers or tones. Even non-musicians have a so-called "piano in the head", where notes are arranged from left to right, low to high, on a horizontal axis. This neurological trick can thus be applied in a spatial neglect intervention method [54,55].

For these aforementioned reasons, musical practice is considered a promising tool for the rehabilitation of USN [49]. However, to date, only a few studies exist on the use of MNT [17,56–58]. These studies use a horizontally aligned instrument, such as a piano, with the objective to increase the active exploration towards the neglected side of the patient. Overall, results were positive, observing significant improvements in the visual exploration of patients, with both short- and long-term effects. One case studied the influence of the congruence of the sound feedback design on the performance of the patient, by comparing multiple sound feedback options, such as congruent sound feedback, i.e., the pitches of the notes are horizontally aligned similar to the keys on a piano, random sound feedback and no feedback, i.e., silence [17]. The researchers observed that the spatial exploration of patients with left-sided neglect was superior in the case of congruent sound feedback and that the use of a musical scale may trigger the preserved auditory and spatial multi-sensory representation of successive sounds.

### 2.5. VR for Rehabilitation

In Virtual Reality, the user is placed in a completely computer-generated three-dimensional simulation, where they can look around and interact with their environment using devices such as a Head Mounted Display (HMD) and gloves or controllers, to create an immersive experience. VR has long since found its place in the world of gaming and entertainment. Nevertheless, VR is increasing in popularity in a clinical setting. A tool for the rehabilitation of children with dystonic cerebral palsy [59], limb pain alleviation [60] and VR exposure therapy for anxiety disorders [61] are just a few examples of the clinical applications of VR [62–67]. Other mixed reality tools, such as Augmented Reality (AR), have also found their way into clinical applications, e.g., neurosurgery [68,69].

VR has many advantages over traditional rehabilitation techniques. First, VR systems can augment and manipulate the interaction of the user with the environment and can quantitatively monitor it, promoting the functional recovery of the patient [70]. Second, opposed to traditional treatment options, VR is modular, enabling the tailoring of the treatment to the patient's needs and physical or cognitive capabilities. Third, there is no need for a clinician as the system can monitor the performance of the patient and quantify how well a task is being performed and give feedback accordingly. This consequently increases the potential for at-home rehabilitation. Lastly, the virtual environment is often

more immersive and enjoyable than conventional treatments, which leads to the patient more actively participating and an increased positive attitude towards the treatment, which increases the overall quality of life of the patient [71].

However, the use of VR introduces several disadvantages as well. First, the immersion aspect can provoke cyber sickness in some, leading to the incompletion of the treatment or an unenjoyable experience. Nevertheless, a large portion of these side effects can be managed by taking appropriate precautions, such as excluding patients prone to motion sickness and following post-immersion procedures [72]. Second, there are studies investigating the spatial perception of distance [73,74] and orientation [75] in VR environments. These studies have found that subjects perceive distances differently in VR than in real life, while the perception of orientation remains the same in VR. This should be carefully considered in therapy use cases where the perception of the user is important. Third, the design of a new VR tool for a clinical setting should always be based on an informed theoretical foundation [76]. Finally, although the cost of VR devices has decreased in recent years, the initial equipment cost is still far greater than traditional pen-and-paper tools, which can be a barrier to their adaption in a clinical setting, especially in lower-income countries [77,78].

As mentioned, VR is becoming more and more present in healthcare applications. On of these applications is the rehabilitation of spatial neglect [79–83]. As there is currently no gold standard for the rehabilitation of USN, new VR interventions can flourish. Moreover, VR provides the opportunity to address and support the multi-sensory characteristics of the syndrome as it incorporates not only visual but also other sensory stimuli. Tactile feedback can be provided using haptic feedback through controllers, and the headphones integrated in a HMD provide auditory stimuli. Moreover, three-dimensional sound can be simulated such that sound is linked to a certain location in space. The use of VR in rehabilitation for USN can hereby accommodate the demand for multi-sensory intervention techniques. Next, clear visual and auditory progress markers motivate the user and help the rehabilitation process. Furthermore, the use of VR provides more enjoyment of the therapy and increases overall motivation. Patients that have suffered from a stroke experience often suffer from post-stroke depressions, originating from the feeling of loss of independence [37]. Self-administering rehabilitation can restore the sense of self-ownership by enabling patients to carry out the rehabilitation independently. Finally, the option for at-home rehabilitation is beneficial for those with mobility issues, often the case for post-stroke patients [84,85].

A variety of studies regarding the use of VR for the rehabilitation of USN exist, such as the grabbing and placing of sushi on a virtual sushi bar [79], following a virtual moving ball with their eyes [80], a rehabilitation tool that offers extra-personal and peripersonal training by performing several tasks in the VR environment [81], a virtual translation of the CBS [82] and a virtual environment based on the PA method, where the HMD emulates the prism [83]. However, no studies have been found that combine music therapy and VR into a rehabilitation tool for USN. Nonetheless, studies combining musical therapy and VR exist, such as a VR system to train everyday hand movements using musical exercises for post-stroke patients [86] or a virtual musical theatre for people suffering from Alzheimer's disease, where patients are immersed in a musical experience among the audience [87].

## 3. Vision and Objectives

The previous section indicates a lack of therapy tools for USN that include multiple modalities, despite the consideration of USN as a multi-sensory deficit. MNT exploits these multiple modalities and is therefore considered to be a promising tool for the rehabilitation of USN patients [49]. The use of VR for the rehabilitation of USN offers the advantage of not limiting the therapy to visual stimuli due to the easy integration of auditory, tactile and motor stimuli. A VR environment facilitates the creation of an immersive experience that can be adapted to the needs of the patient. Moreover, the use of a digital tool, as opposed to manual assessment, provides the opportunity for more unbiased data collection during

therapy. Information, such as head direction or responses to stimuli in the VR environment, can be tracked to evaluate progression over time.

The objective of this research is to investigate whether MNT for USN can be successfully translated to an immersive VR environment and whether the data collected during the immersive training can be used for the assessment and thus evaluation of the progress of patients affected by USN. To achieve this, a digital tool has been designed that includes an immersive VR environment in which the patient has to perform several musical exercises, tracking and displaying session data on a therapist dashboard.

To ensure that this tool achieves its objective, expert knowledge about the needs of the target audience and offered therapy is necessary. An iterative participatory design approach has been taken, involving experts and patients in the design process. This research has been established in close co-operation with specialists and therapists from the Rehabilitation Department of the Ghent University Hospital. To evaluate the resulting system, a pilot study, approved by the ethics committee, has been conducted involving several patients who were currently undergoing treatment for USN at the Rehabilitation Department of the Ghent University Hospital.

Three research questions have been investigated:

1.　Can MNT be effectively translated into a VR environment?
2.　Can the newly designed VR system be used for the treatment of adults with USN with an acquired brain injury?
3.　Will the spatial performance of the patient with an acquired brain injury evolve positively after treatment with the VR system?

## 4. Design of the VR Therapy for USN

The multi-sensory VR rehabilitation tool enables the user to perform several musical exercises on a virtual xylophone, while incorporating three sensory modalities, namely auditory, visual and motor. During these exercises, patients are triggered to explore their neglected side.

To ensure that a therapist can guide and follow an ongoing session, the designed platform consists of two modules, one for the patient and one for the therapist, as shown in Figure 5. For the therapist, there is a dashboard, accessible on a computer, to oversee and control the therapy as it is in progress. For the second module, the VR therapy, the patient has to wear a HMD and use two controllers, representing mallets, to stroke the keys of the virtual xylophone.

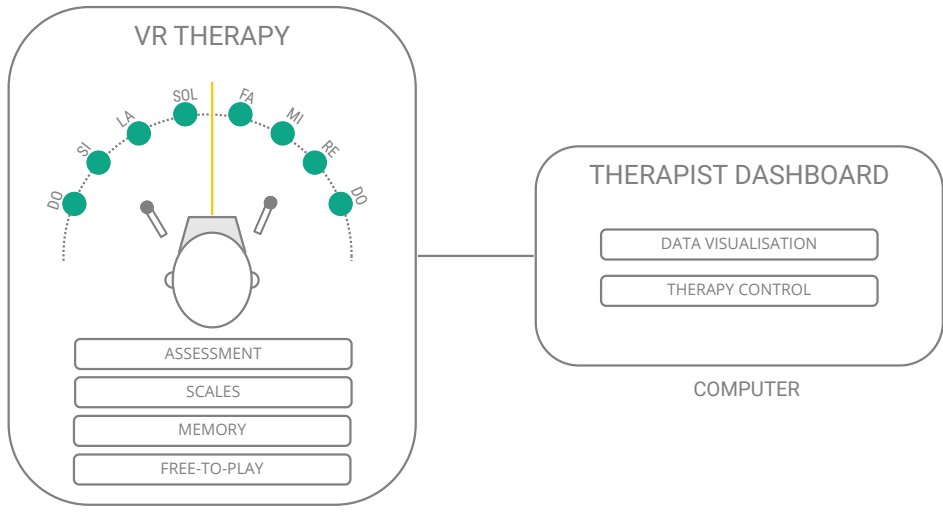

**Figure 5.** The designed VR therapy tool for USN consists of two modules: the VR therapy module, for the patient, who has to wear a HMD and use controllers, and the dashboard, accessible on a computer, where the therapist can control and oversee the session.

The following sections will first elaborate on the designed VR therapy module and its four tasks, followed by an explanation of the designed therapist dashboard.

### 4.1. VR Therapy Module

During a VR therapy session, the patient will see an arrangement of several musical keys, similar to a xylophone, aligned in a semi-circle around them, as shown in Figure 5. Using the controllers, notes can be played by hitting a key with a percussion mallet, i.e., the motor modality of the therapy. The notes produced by the keystrokes are spatial sounds, i.e., originating from their location in space, where its pitch also depends on its location in the virtual environment, i.e., the auditory modality. Finally, when hit, the keys light up in a different colour for approximately 1 s before turning back to their original colour, corresponding to the visual modality.

The VR therapy module contains four tasks, called Assessment, Scales, Memory and Free-to-Play. The Assessment task aims to assess the severity of the spatial neglect of the patient and their progress in the rehabilitation process. The other three tasks aim to change the behaviour of the patient by training them to respond to triggers in their environment, a top-down approach.

The use of a VR environment enables us to adapt the setup of the xylophone to the current task and difficulty level. More specifically, the number of keys of the instrument and their location in space can instantly be changed to meet therapy needs. The location of the keys changes based on the objective of the task. Since the *Assessment* task aims to assess progress, the keys will be symmetrically spaced on the semi-circle, as shown in Figure 6. However, for the three training tasks, the keys will be located off-centre towards the neglected side of the patients' environment, as shown in Figure 7, a bottom-up approach. More specifically, two keys will always be located on the ipsilesional side, while the other keys are situated in the neglected side of the environment. In this setup, the key at the extremity of the instrument is an extra key, producing a special sound. This modification is necessary as people with USN benefit from these anchor points in their neglected side of space. Moreover, a red target will occasionally appear during these training tasks. The users are required to look directly at this target for a few seconds, before continuing with the task. This addition is necessary to keep the patient aware of the location of their mid-line. Finally, the difficulty level of each task is reflected in the number of keys shown, namely difficulty 1 to 4, which corresponds to 6 to 9 keys, respectively. Each task, except the *Assessment*, consists of several of these levels, and thus a different number of keys based on the set difficulty level, as shown in Figure 8.

Next, each task will be discussed in more detail.

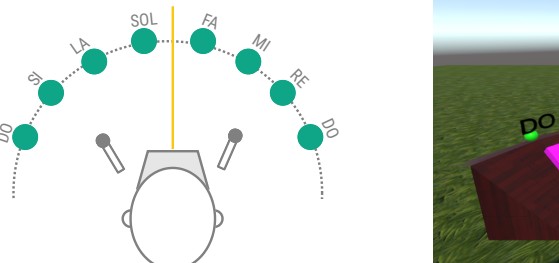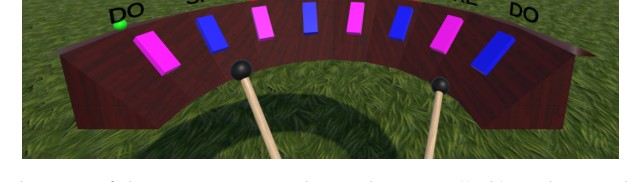

**Figure 6.** General setup of the symmetrical setup of the *Assessment* task in schematic (**left**) and virtual environment (**right**).

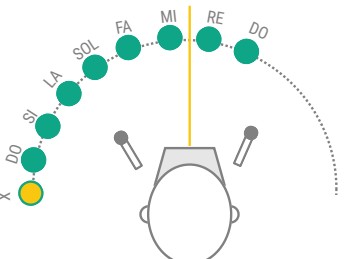
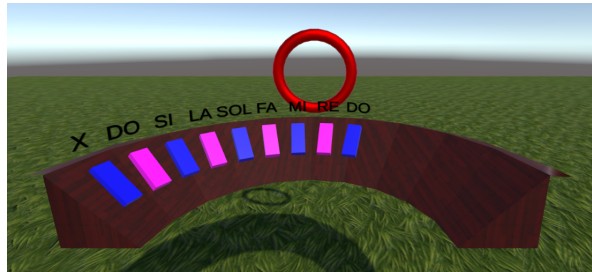

**Figure 7.** General setup of the asymmetrical setup of the three rehabilitation tasks in schematic (**left**) and virtual environment (**right**).

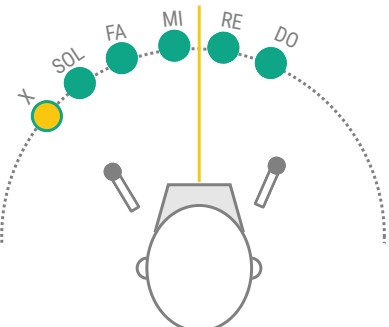
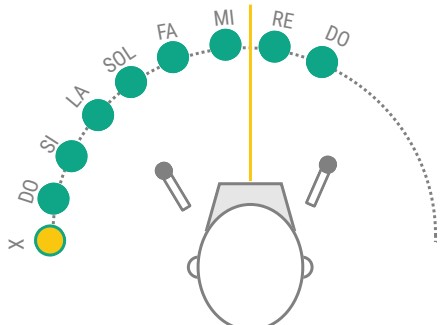

**Figure 8.** Setup of instrument during the rehabilitation tasks. Difficulty One (**left**) and difficulty Four (**right**).

### 4.1.1. Assessment Task

The *Assessment* task is the only task that does not use difficulty levels and should not be played each session. The objective of this task is to assess the progress of the patient during their rehabilitation and can be performed whenever the therapist deems it necessary. For this task, the instrument keys are set up symmetrically (cf. Figure 6), regardless of their neglected side, assuming that the head direction of a patient with USN will indicate a bias towards their non-neglected side.

During this task, the patient is asked to freely play the instrument. However, test runs and feedback from the experts indicated the need to incentivize patients to continue playing different notes. Therefore, each key is supported by a small ball, that starts up in the air and slowly falls towards the keys. Patients are instructed to avoid letting the balls touch the keys. When a ball is close to the surface of a key, it turns green, indicating that the patient can then hit the corresponding key to send the ball back up into the air. When the ball touches the key, it turns red (cf. Figure 6 right). This gamified element triggers the active exploration of their entire environment during the *Assessment*.

### 4.1.2. Scales Task

The *Scales* task is the first training task. In this task, the patient has to play scales on the xylophone, i.e., the patient has to sequentially hit each key in order, starting from their non-neglected side towards their neglected side. For left-sided neglect, this corresponds to playing the keys from right to left. This task familiarises the user with the off-centre layout of the instrument (cf. Figure 7) and explores whether the patient has a certain threshold for which they believe it is the last key on the instrument.

### 4.1.3. Memory Task

The *Memory* task asks the patient to memorise a sequence of notes that is shown to them and then repeat this sequence. This task starts from the lowest difficulty level, showing a sequence of one note, each time increasing the difficulty, i.e., the length of the sequence or the number of keys on the virtual xylophone. When the patient plays an incorrect key, the sequence is shown again; when they play the correct key, positive

feedback is given. This process is repeated with increasing difficulty until the patient errs too often.

The cues that are given to indicate the sequence to be memorised are the same as when a key is hit, i.e., the key changes colour and an animation is played, accompanied by the spatial sound of the note.

The *Memory* task urges the patient to actively explore their entire environment and respond to both visual and auditory stimuli. Since the entire instrument is not visible in one glance, the patient has to move their head to locate and react to the spatial audio. This thus aims to improve the ability to respond to external multi-sensory triggers, located in the contralesional side of space.

### 4.1.4. Free-to-Play Task

The final task is the *Free-to-play* task. In this task, the objective was to provide the patient with some 'cognitive downtime' by letting them play freely. However, similar to the *Assessment* task, to motivate the patients to keep on playing during this task, the same game mechanics have been included. The use of small balls falling to the surface of the keys again triggers the active exploration of the patient, urging them to explore their entire environment to avoid any balls hitting the keys.

### 4.2. Therapist Dashboard

The second module is the therapist dashboard, aimed to oversee and control the ongoing session. The dashboard shows the head direction of the patient over time and the keys that are played during a session, as shown in Figure 9. Using the head direction information of previous sessions, the therapist can see if improvements are noticeable and the patient gives more attention to their neglected side. Via the dashboard, the therapist can start or pause the therapy and select which task at which difficulty level is started next.

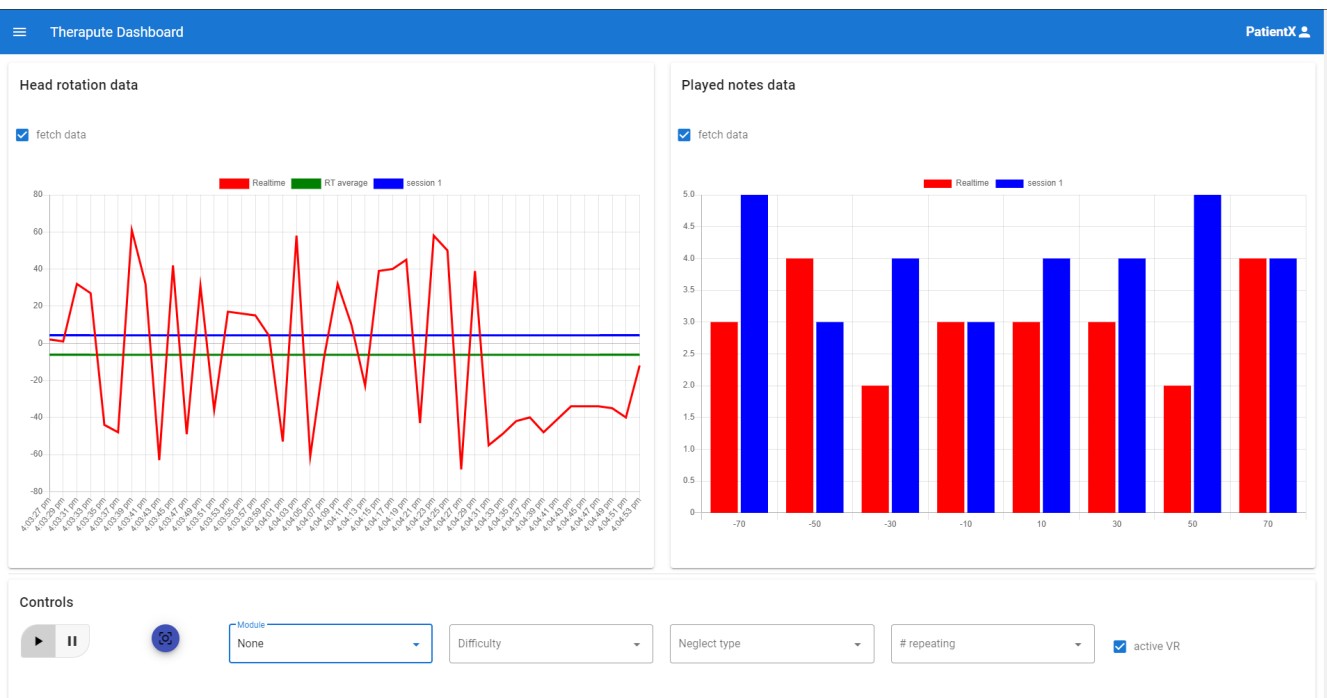

**Figure 9.** Therapist dashboard to control and oversee the therapy.

## 5. Implementation Details

Based on the functional requirements presented in Section 4, a working prototype has been developed for the pilot study. However, as the resulting system is used in a research track, some additional non-functional requirements need to be considered. Specifically,

the modifiability of the system is quite essential. Firstly, the system's dashboard needs to be interchangeable with different implementations. Future research might focus on different dashboard designs and their usability, which requires new implementations. Secondly, the system's data storage needs to be as flexible as possible. Legal, ethical or GDPR-related issues might require changes to the data storage deployment, for which it might be necessary to opt for either a local or server-based solution. Furthermore, future research will consider integrating the system with existing data stores—for example, for patient data records.

The remainder of this section discusses the decomposition of the system into individual components connected by interfaces between them. This is followed by an overview of the technologies used and, finally, a discussion of the deployment of the test setup.

### 5.1. Design Overview

Based on the functional and non-functional requirements, the system is decomposed into three separate components. Figure 10 shows the system's decomposition into a *VR system*, a *data storage* component and a *dashboard* component. The *VR system* is responsible for managing and displaying the VR environments to the patient and collecting the data from the therapy sessions and the HMD. The *data storage* component stores and delivers the data displayed on the dashboard. The *dashboard* component is a stand-alone dashboard application that requests the data from the *data storage* to display them in graphs on the screen. The *dashboard* also issues control requests to the *VR system* to control the therapy and configure the exercise.

### 5.2. Interfaces

In order to allow easy modifiability with little constraint on the *data storage* and *dashboard* components, the technologies for the interfaces between the components are based on the TCP/IP protocol stack. Figure 10 shows the interfaces between the identified components. Both interfaces between the *VR system* and the *dashboard* and the *VR system* and the *data storage* use a network-based message queue. The communication between the *dashboard* and the *data storage* happens over the database's TCP/IP communication protocol.

### 5.3. Technologies

The *VR system* component was entirely developed in Unity3D (https://www.unity.com/, accessed on 19 December 2021). This is a game engine with broad support for VR development. The data store is a PostgreSQL database (https://www.postgresql.org/, accessed on 19 December 2021) with a Python script to handle the communication with the *VR system*. The *dashboard* is a vue.js application. For the message bus, ZeroMQ is used (https://zeromq.org, accessed on 19 December 2021). It is an open-source universal messaging library with support for TCP transport. Finally, the HMD is an Oculus Quest 2 (https://www.oculus.com/quest-2/, accessed on 19 December 2021) connected with a link cable to a gaming laptop featuring an off-the-shelf VR-ready GPU. The technologies used in each component are indicted in Figure 10.

### 5.4. Deployment

For the implementation of the proof of concept used in this study, all components are deployed on a single machine. There are two primary motivators for this. Firstly, security is needed due to the sensitive nature of the data. Secondly, the availability of the system is required due to the sensitive nature of the therapy. The most affordable tactic to achieve both is restricting the entire software stack to one system and eliminating all external communication. Therefore, the implementation also needs no internet connection. The only external hardware required is the HMD and the controllers connected to the PC, which directly communicate with the *VR system*.

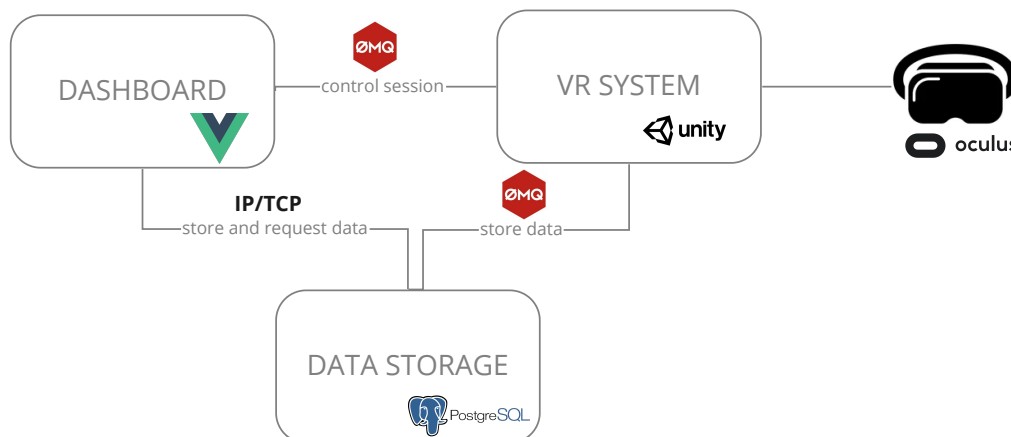

**Figure 10.** The architecture of the designed VR platform.

## 6. Evaluation Set-Up

This research aimed to evaluate the proposed VR therapy in practice. Therefore, therapy sessions were performed with diagnosed USN patients. A control group of users without USN also performed the therapy exercises using the VR system. This study was performed at the Ghent University Hospital with clinical patients and the IDLab offices with non-clinical users. Ethical approval was obtained for the study and all participants signed an informed consent form before participating. The study was approved by the Medical Ethics Committee of Ghent University Hospital on 19 March 2021 [B6702021000166].

This section will discuss the test methodology that was adopted, and the timeline of the pilot study. An overview of the participating individuals and the recruitment process is given. Finally, the data collection during the study is discussed.

### 6.1. Method

The clinically diagnosed patients with USN received VR-based therapy for two weeks as part of their rehabilitation program at the hospital. During the two-week period, sessions took place on Monday, Wednesday and Friday in the presence of a therapist. Each session had a duration of approximately 30 min, in which, first, the *Assessment* took place, followed by the *Scales*, *Memory* and *Free-to-Play* tasks. On four occasions during these two weeks, the *Assessment* task of the therapy tool was performed. During each session, all other VR rehabilitation tasks were performed. Figure 11 shows a timeline of the two-week test period, on which the tasks for each session are indicated. At the end of the two-week period, an interview with the patients collected additional feedback and impressions.

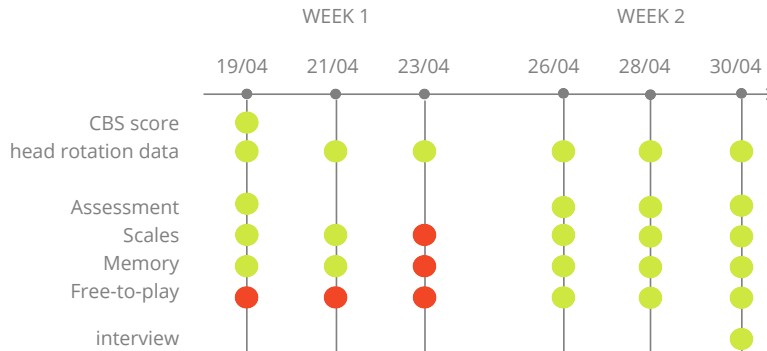

**Figure 11.** The timeline of the clinical study with patients indicates 6 sessions during 2 weeks. Each session consisted of a few tasks and data collection. Green markers indicate the available data; red markers indicate unavailable data.

As a baseline to compare the performance of the clinical patients with, the same therapy sessions were performed with test users without USN. These test users would each perform a single therapy session of around 20 min. The test conditions mimicked as closely as possible the pilot study with the patients. However, these tests were performed at our computer science lab instead of the hospital.

### 6.2. Recruitment and Participants

The Rehabilitation Centre of the Ghent University Hospital selected four participants for this study. These 4 patients were undergoing rehabilitation therapy for USN at the moment of recruitment and all patients received their first VR therapy session on 19th April 2021 and their last session on 30th April 2021. The participants were all older than 12 years, suffered from an acquired brain injury with USN and were at least six months post-injury. Of the four participating patients, three had left-sided neglect, and one had right-sided neglect. Furthermore, every patient was assessed with standardised USN assessment tests prior to the study. These assessments showed that Patients 1 and 3 had significantly more severe neglect than Patients 2 and 4. Moreover, Patient 4 had Apraxia, i.e., a neurological disorder characterised by the inability to perform familiar movements on command, and a language disorder.

The leading researchers recruited the non-diagnosed test users of this study themselves. In total, four test users participated. Each user was also assessed using the CBS to confirm that they did not have USN. The TAP neglect test was not performed on these test users.

The information for each participant is summarised in Table 1. The first column is the pseudonym of the individual used in this paper. The second column indicates the participant's side of the neglect. The third and fourth columns present the CBS and TAP score. For Patient 4 and all test users, the results of the TAP assessment are not available.

**Table 1.** A table summarising the neglect assessments executed by the clinic. A higher CBS score indicates more severe neglect. More omissions and a higher reaction time on one side in the TAP test indicate neglect towards that same side.

| Users | Neglect | CBS Score | TAP | |
|-------|---------|-----------|-----|-----|
| | | | Left | Right |
| Patient 1 | left | 19/30 | reaction: 481 ms omissions: 15/22 | reaction: 719 ms omissions: 2/22 |
| Patient 2 | left | 8/30 | reaction: 1074 ms omissions: 5/22 | reaction: 834 ms omissions: 1/22 |
| Patient 3 | left | 19/30 | reaction: 1234 ms omissions: 18/22 | reaction: 799 ms omissions: 2/22 |
| Patient 4 | right | 7/30 | – | – |
| Test user 5 | none | 0/30 | – | – |
| Test user 6 | none | 0/30 | – | – |
| Test user 7 | none | 0/30 | – | – |
| Test user 8 | none | 0/30 | – | – |

### 6.3. Data Collection

The CBS and TAP scores, as discussed above, were collected at the start of the study. During the VR therapy, data about the user's interactions with the system were recorded. Specifically, the head direction of the user and the played notes were registered in the VR application. Additionally, the time between each played note was also measured during the *Scales* task. For the *Memory* task, the instruction sequence, each played key and whether the key was correct were saved. The *Free-to-Play* task also recorded how long each ball rested on the surface of the key before it was hit. Finally, after all the VR therapy sessions were finished, the patients were interviewed to collect their impressions of the system. The medical experts were also interviewed to learn their findings of the proposed system and therapy. Figure 11 indicates the collected data for each session in the clinical pilot study.

The green markers indicate the performed tasks for which data were available. The red markers show for which performed tasks data were not available. During the tests with test users, all the above mentioned data were collected for each user.

## 7. Results

The following section presents all results of the two-week pilot study with patients and the tests with non-clinical test users. All data that include a Euclidean angle have been standardised as follows. A positive angle means a rotation towards the neglected side of the patient, regardless of the neglect type of the patient. This allows the comparison of patients with left and right neglect in an equal manner. Furthermore, whenever applicable, the names of the keys and their positions are indicated on the axis of the graph, i.e., the angle between the note and the centre of the VR environment. However, the name of the key is less significant than the position with respect to its impact on USN.

### 7.1. Assessment Task

During the *Assessment*, the head direction of the users is recorded, resulting in traces for each user of each session. The average of all traces is calculated and compared for patients and test users. Figure 12 shows the first 60 s of the aggregations. The patients show a consistent bias of 20 to 30 degrees towards the non-neglected side during the first 11 s of each *Assessment*. After these 11 s, it appears that the average resides much closer to the centre of the playing area, albeit with a more significant standard deviation. This means that there is a large consensus among all patients to look at the non-neglected side during the first 11 s. The test users do not show this behaviour. Their average head direction is close to 0 degrees throughout the entire session. The standard deviation for the test users remains relatively consistent and is similar to that of the patients after the first 11 s. Figure 12 also indicates that, on average, the first key was played after approximately 11 s following the beginning of the traces.

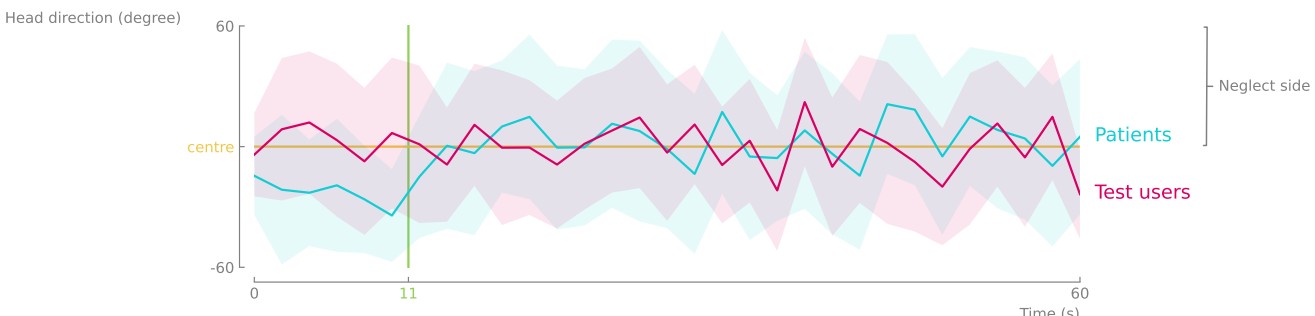

**Figure 12.** The average head direction of the first 60 s of all the *Assessment* tasks is aggregated and plotted for the clinical patients (blue) and the non-clinical test users (pink). The patients indicate a bias of around 20 degrees towards their non-neglected side for the first 11 s, the average time until the first key is played (green). This bias is corrected after this first stimulus, resulting in an average head direction fluctuating around the middle for the remainder of the 60 s.

The bias of the head direction during the *Assessment* task is further investigated. As the purpose of the *Assessment* task is to assess the severity of the USN in a patient, it is interesting to compare it to the CBS scores of the patients. Figure 13 shows how the CBS score correlates to the average head direction during the first 11 s (left) and the remainder of the task (right). Participants with a CBS score close to zero consistently have an average head direction close to zero. As the CBS score increases, bias in the average head direction can be observed. During the first 11 s, all patients with CBS > 0 have a bias towards the non-neglected side. For the remainder of the task, patients with CBS <= 8 show no bias any more, and they position their head in the centre, while patients with CBS > 8 show a bias towards the neglected side.

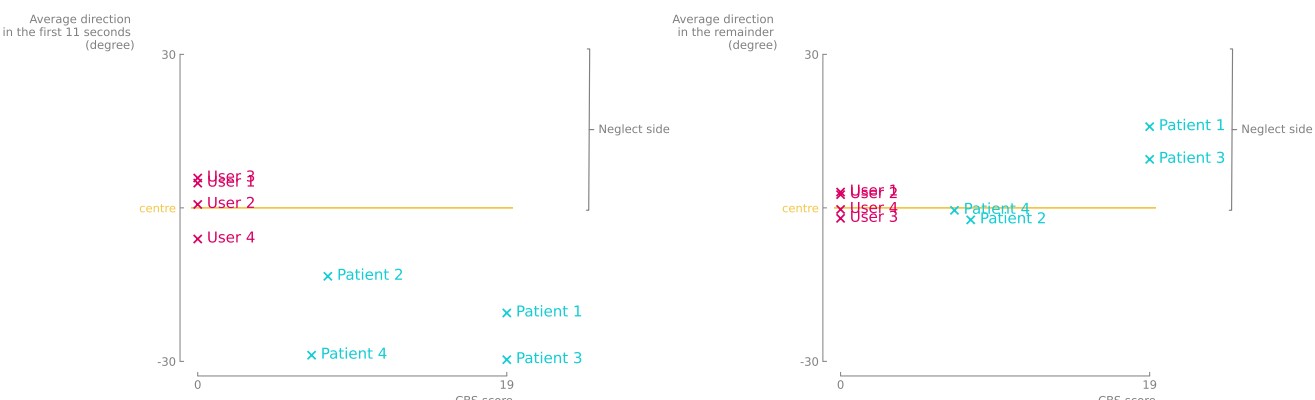

**Figure 13.** (**Left**) Users with a higher CBS score indicate a higher bias towards their non-neglected side compared to the users with a CBS score of 0. Within the patient sample, no significant difference in head direction can be observed based on the CBS score for the first 11 s. (**Right**) The CBS scores of all participants are plotted against the average head direction during the remainder of the *Assessment*. These show that the bias of the head direction has shifted towards the non-neglected side for users with a higher CBS score.

However, when observing the difference between the averages during the first 11 s of the *Assessment* and the rest of the task, a more significant correlation with the CBS score becomes apparent. In Figure 14, this difference is calculated and plotted against the CBS score. Indeed, the correlation appears linear within the range of CBS scores of 0 to 19. More specifically, users with a CBS score of 0 show minimal differences during the first 11 s and the remainder of the *Assessment* task. The larger the CBS score, the more significant this difference in bias becomes. The patients appear to apply a correction to their head direction in order to observe more of the environment.

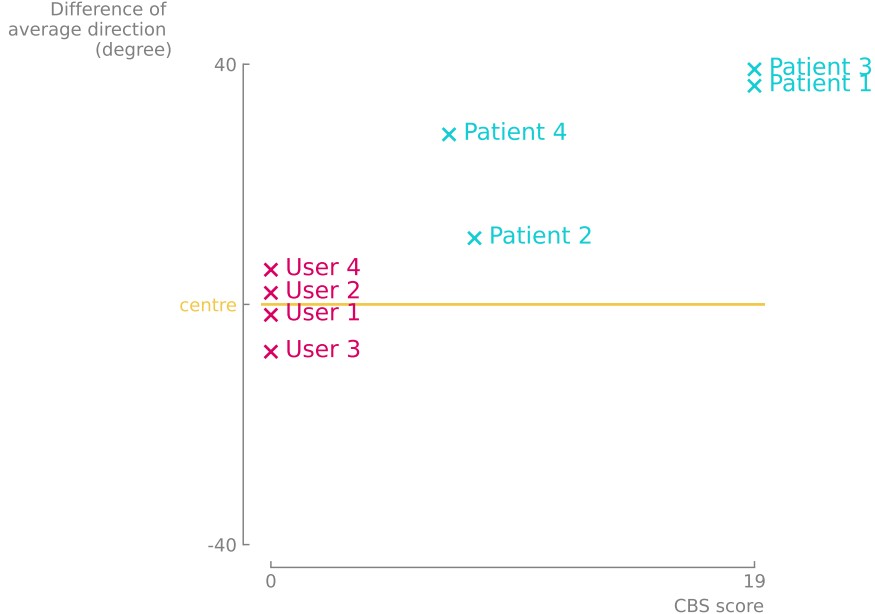

**Figure 14.** The CBS scores of all participants are plotted against the difference between average head direction during the first 11 s and the remainder of the *Assessment*. These indicate a linear correlation with a bias towards the neglected side. The higher the CBS score, the greater the bias towards the neglected side of space.

### 7.2. Scales Task

The time to play the following key in the scale is extracted during the *Scales* task. Figure 15 shows the aggregated values of all sessions for patients and test users. The data in the graphs are the median time it took to play each key for all sessions, regardless of the difficulty level and the Q1 and Q3 quartiles. The time to play the first note on the scale is omitted in these figures. The value of this first keystroke represents the reaction time to the presentation of the visual stimulus on the first key, while all other values represent the time in between two keys being played. Leaving out the first key allows for a better comparison of the data. Patient 1 had difficulties with the last note in the sequence. It took them significantly longer to play that note than any other note. Moreover, the spread of the time values for this key is significantly wider; this suggests that they were not consistent in the time to play this key. For all patients, except Patient 1, the time to play the following key on the scale gradually reduces as they move further towards the neglected side. The time needed by the test user to play each key is significantly lower than that of the patients. In addition, the spread of the time values is much narrower for the test users compared to the patients.

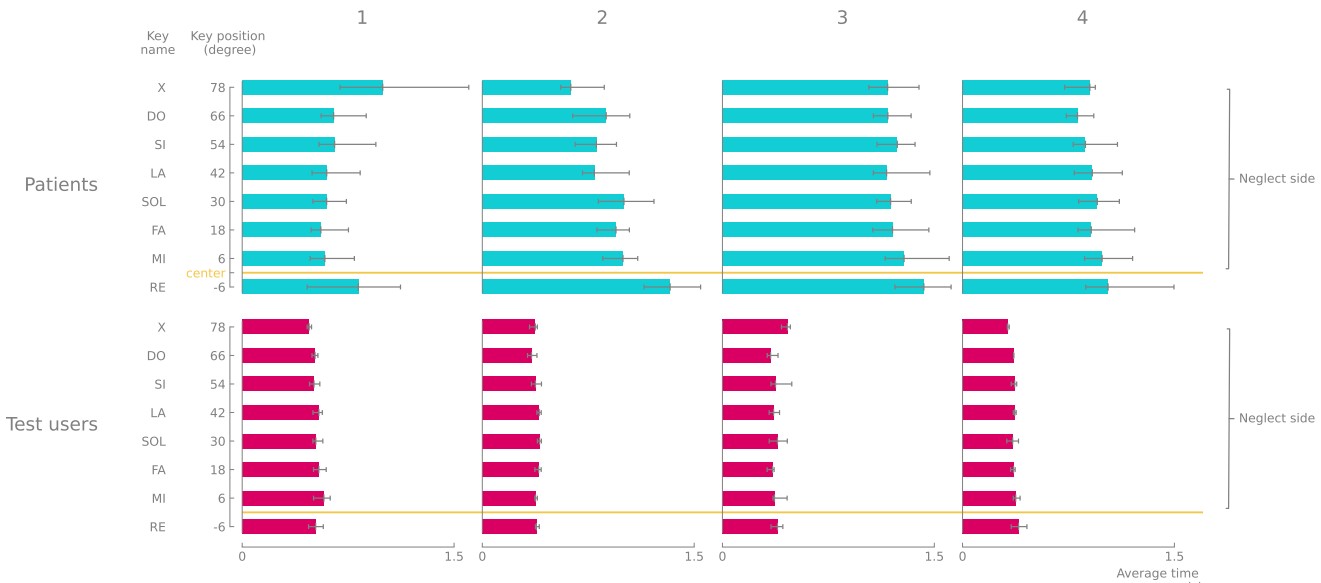

**Figure 15.** The median of the time it took to hit the required note during the *Scales* task for patients and test users. Quartiles Q1 and Q3 are indicated by the horizontal lines. Patient 1 required more time to hit the final key at the extreme end of their neglected side, whereas, for all patients, the time to hit the next key reduced gradually as they moved along the scale. The test users indicated a significantly faster response time compared to the clinical users.

### 7.3. Memory Task

For the *Memory* task, the correctly played keys and the mistakes are recorded. Firstly, the success rate of correctly repeating a sequence of notes is shown in Table 2 for all participants. This measure represents the fraction of correctly played sequences of all sequences played, including every retry of a sequence until they succeeded. In general, the non-clinical test users made fewer mistakes across the board than the clinical patients. Moreover, there are significant differences in success rates between the different patients. Patient 2 played the most erroneous sequences, while Patient 4 had a success rate approaching that of the non-clinical test users. Test user 4 did not make any mistakes.

**Table 2.** The success rate for replaying sequences correctly during the *Memory* task for each patient. Patient 2 had the lowest success rate, while the success rate of patient 4 approached that of the non-clinical users.

| User | Success Rate |
|---|---|
| Patient 1 | 0.62 |
| Patient 2 | 0.35 |
| Patient 3 | 0.60 |
| Patient 4 | 0.85 |
| Test user 1 | 0.83 |
| Test user 2 | 0.92 |
| Test user 3 | 0.94 |
| Test user 4 | 1.00 |

Figure 16 shows the distributions of the notes in the instruction sequences and the played sequences for patients and test users. The size of the marker indicates the relative frequency of that note in both distributions. For all participants, most keys were played correctly, which can be seen by the large values on the diagonals of the graphs. However, the patients made significantly more mistakes than the test users. In particular, Patient 2 made many mistakes on individual keystrokes. Remarkably, Patient 2 confused almost each key for every other key at least once. A higher concentration can be seen towards the neglected side. This indicates that keys in the instruction sequence on the neglected side were often confused for another key. The data of Patient 3 show a similar pattern, where some keys in the far neglected side are more confused for other keys. The mistakes made by the other users are situated more closely to the instructed keys. In other words, the mistakes by the other users meant that they played a key only a few keys apart from the instructed key. The test users made minimal mistakes, which resulted in a sparse plot.

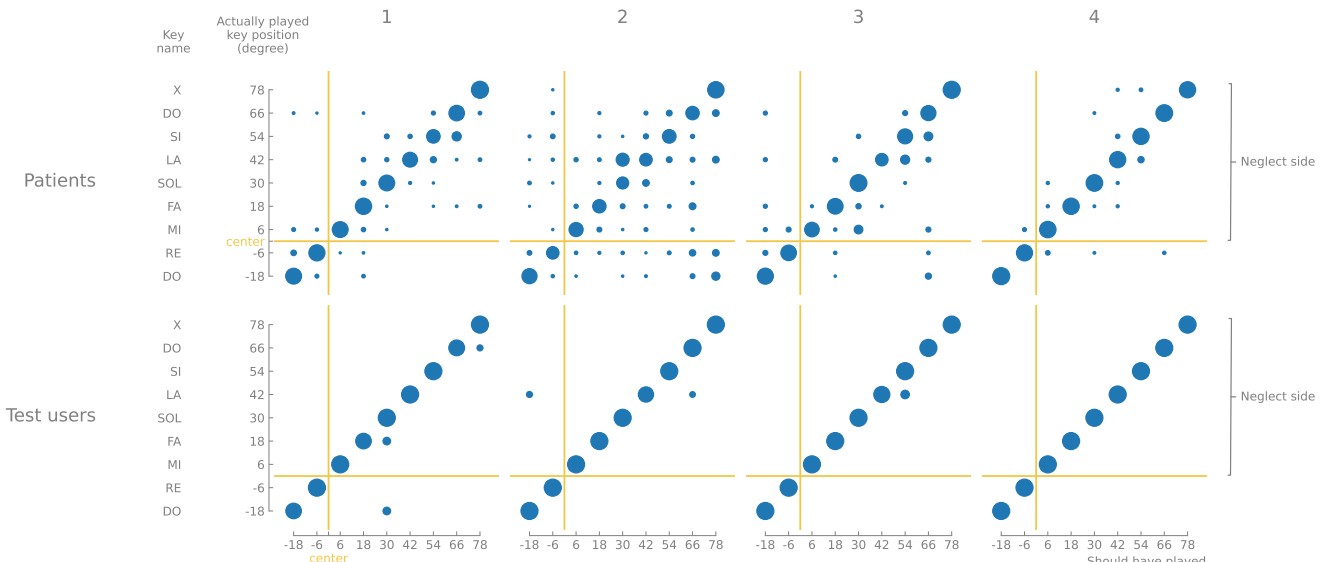

**Figure 16.** The distribution of the notes in the example sequence. The keys that should have been played, the played sequences and the keys that were played indicate that, overall, most keys were played correctly. Nonetheless, patients indicated a higher concentration of mistakes in general, mostly situated in their neglected side.

### 7.4. Free-to-Play Task

During the *Free-to-Play* task, the time for which each ball was resting on the surface was measured. Figure 17 presents these data for patients and test users. The data are the median time each key was not played while it should be played, as well as the Q1 and Q3 quartiles to indicate the spread of the timing values. For Patients 1, 2 and 3, an increase

in time could be seen for the keys on the far most neglected side. Patients 2 and 4 also showed a significant increase for some keys in the centre of the playing area. Furthermore, in Patients 1 and 3, it was most apparent how keys up until 30 degrees toward the neglected side were played the quickest, while, beyond this angle, they were left unplayed for longer. Moreover, the spread of the timing values increased the further the key was to the neglected side. Finally, the test users left far fewer keys unplayed, and when they did, the duration was always considerably lower compared to the patients.

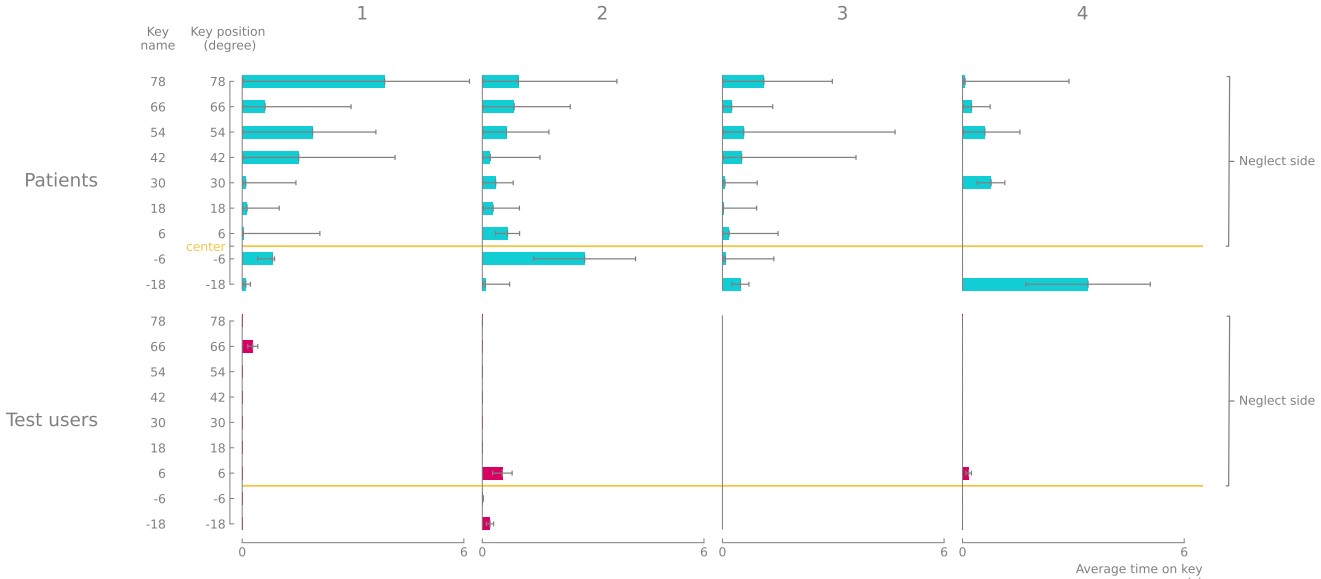

**Figure 17.** The median of the time for which a ball was lying on the surface for each key during the *Free-to-Play* task is remarkably higher for the patients than for the test users. Moreover, the spread of the values is wider, as seen by the quartiles Q1 and Q3, indicated by the horizontal lines. For most patients, keys on the extremity of the instrument in their neglected side remained unplayed for the longest. Patients 2 and 4 showed a somewhat similar increase for keys in the middle.

*7.5. Evolution*

Because the purpose of the rehabilitation tasks was to motivate patients to explore the entire playing area and pay attention to their neglected side, it was reasonable to investigate whether any evolution was noticeable across the two-week period. In particular, the *Memory* and *Free-to-Play* tasks were most interesting to investigate as these two tasks motivated the patient to explore the entire environment to notice stimuli and respond to them. The tasks themselves did not force the participant to look in a particular direction. Figure 18 plots the evolution of average head direction for the *Memory* and *Free-to-Play* tasks. The data of all six sessions are used. A slight gradual transition of the average head direction is visible across the two-week period for all patients.

*7.6. Interviews*

The feedback from the questioned patients suggests that they perceived the therapy as neither too complex nor too simple. Nevertheless, according to the participants, a high concentration level was required, which sometimes made it tiring. Two patients mentioned that their performance on the different tasks improved during the two weeks. One patient anecdotally said that now she was more inclined to look towards her neglected side while sitting in the car. However, the other two patients mentioned no noticeable differences in real life. Finally, all three questioned patients would be willing to receive this therapy as part of their current treatment.

The medically trained therapists that supervised the VR sessions and who administered regular therapy believed that the proposed therapy tool could treat neglect at an

early rehabilitation stage. Training three modalities, visual, auditory and motor, posed a significant advantage for using VR in USN therapy. Moreover, the increased independence and immersion of the therapy were considered beneficial. Motion sickness, epilepsy and the inability to visually explain things were reported as disadvantages of VR in USN therapy. Lastly, it is believed that the proposed tool would be a meaningful addition to the other therapy methods of the clinic.

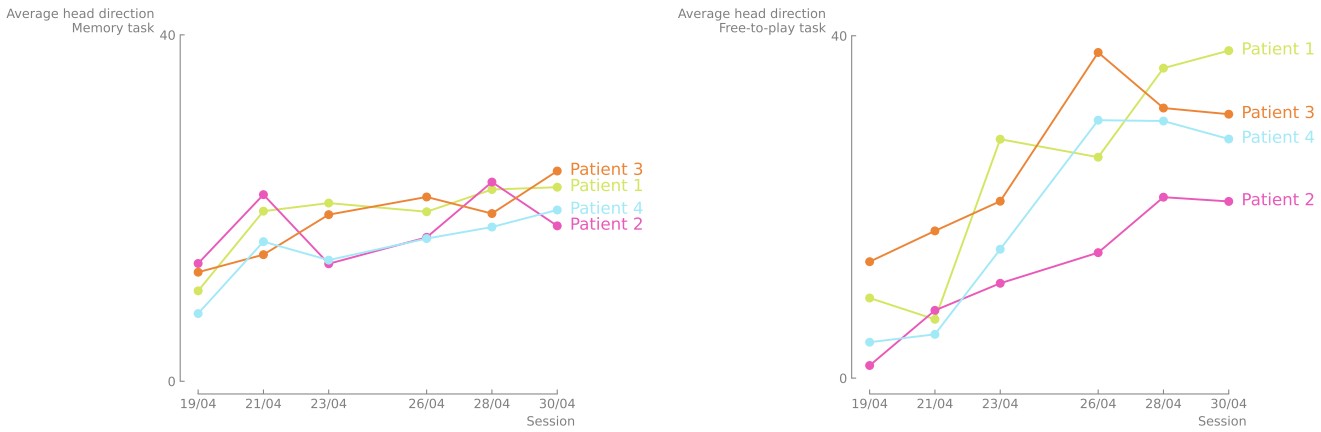

**Figure 18.** (**Left**) The average head direction for the *Memory* task for each patient during all 6 sessions indicates a gradually increasing correction towards their neglected side. (**Right**) The average head direction for the *Free-to-Play* task for each patient during all 6 sessions indicates a gradually increasing correction towards their neglected side.

## 8. Discussion

A thorough discussion of the results and an investigation of the study's limitations are provided in this section. Firstly, the most interesting and notable findings of the different therapy tasks are presented. Next, based on these findings, the research questions are revisited and discussed. Finally, the limitations of the study and the resulting data are investigated and discussed.

### 8.1. Findings

All patients could perform all tasks with sufficient proficiency. However, in all cases, their performance of the tasks was significantly lower than that of the test users. This may indicate the sufficient motor and cognitive challenges that the tasks present. It could also indicate that the tasks do not sufficiently act on the USN, and additional challenges can be added in future iterations. The diagnosed USN did not prohibit any of the patients to perform a task, nor did their diagnosis lead them to completely ignore part of virtual instrument. However, the results of the VR rehabilitation show some patterns that suggest that the USN has an impact on the patients' performance and that they need to adapt their behaviour to perform them successfully. These patterns in behaviour are discussed in detail below, and possible interpretations are presented.

#### 8.1.1. Assessment Task

The data of the *Assessment* show a clear and interesting phenomenon. During the first 11 s of the *Assessment*, all patients are biased on the average head direction towards their non-neglected side. The assumption is that this is their natural behaviour, i.e., the patients tend to look and explore their non-neglected side when no stimuli are presented. However, after approximately 11 s, when, on average, the first key is played, the average head direction shifts towards the centre and sometimes even towards the neglected side. It is assumed that this is taught behaviour of the patients, i.e., whenever external stimuli are presented, they shift their attention to the neglected side. Patients 1 and 3 corrected their head direction more severely compared to Patients 2 and 4. This is most likely related to

the significantly higher CBS scores of Patients 1 and 3. As they had more severe neglect, a more significant correction to their head direction is warranted, to observe the entire environment successfully. This correlation is evident in Figure 14 for the available data. Ultimately, the *Assessment* task should result in a metric that accurately predicts the degree of USN in a patient. Additional data points of more patients would be needed to further investigate this behaviour and the correlation with the CBS score.

### 8.1.2. Scales Task

For the *Scales* task, two patterns are observed. First, for all patients, except Patient 1, the time to play a key gradually decreased the further it was towards the neglected side. Second, Patient 1 needed significantly more time to play the ultimate key. The reduction in the time to play the following key can be explained by the cognitive effort required to play the first key compared to the keys following it. Once the participant has found the pattern, i.e., playing one key after the other, this action can be performed faster and with less cognitive effort. This behaviour can most clearly be seen for Patients 2 and 4. The other patients and the test users also had the same pattern but less pronounced. A possible explanation for the second pattern is that the different nature of the ultimate key impacts the perception of the users on the neglected side. Specifically, the last keys play a sound that is not congruent in pitch compared to the other keys. The patient could be conditioned not to play the last note as it does not belong to the scale. However, this appears to only happen for tasks of the highest difficulty (nine keys). This explanation is in line with the findings of Bernardi et al. [17], which stated that individuals performed better under conditions with congruent sound feedback compared to random sound feedback. In this case, patients needed more time to play the last key, which is the only key to produce a sound not part of the musical scale, i.e., random sound feedback, compared to the rest of the keys, which each produce a pitch part of the musical scale, i.e., congruent sound feedback.

### 8.1.3. Memory Task

In the data of the *Memory* task, there are also two patterns. The first pattern can be seen for all participants. They mistakenly play a key very close to the instructed key, e.g., when the instruction sequence shows an Fa, but the user plays an Mi or a Sol, both adjacent to Fa. This is expected behaviour and can be classified as a consequence of the participants' cognitive capacity, specifically memory. The number of mistakes of this type is relatively low. The second pattern can mainly be seen in Patient 2 and to a lesser extent in Patient 3. These patients make another mistake with a relatively higher occurrence than the former type. When playing a sequence, they make more mistakes when the instruction key is in the most peripheral area of their neglect than closer to the centre, e.g., Patient 2 often plays the first or second key when the eighth or ninth are instructed. Furthermore, Patients 2 and 3 have the lowest success rates for playing a correct sequence. Exploration and reaction to visual and auditory stimuli are crucial in this task. The users can respond to the auditory stimulus to get a general idea of the location of the key, but they usually need to look for the visual cue to observe the exact key to play. The assumption is made that the average head direction during a task is a good indicator of exploration. That is, the greater the deviation of the average head direction from the centre of the playing area, the better the user is exploring that peripheral area. When looking at the evolution of the average head direction during the *Memory* task, a gradual shift can be seen moving away from the centre. This could indicate that the participants perform better exploration as the rehabilitation sessions progress. Thus, it could be taught behaviour from the tasks that persists in between sessions. The test users made very few mistakes in general. Therefore, minimal data on their behaviour are available, and an insightful comparison is infeasible.

### 8.1.4. Free-to-Play Task

In the *Free-to-Play* task, some behaviour is visible in the data of Patients 1, 2 and 3. Specifically, the more towards the neglected side it is, the more likely a key will be left

unplayed for longer. This behaviour is unmistakable evidence of USN. This task has no limited exploration phase as with the *Memory* task; the cues appear continuously and thus exploration is required continuously. An action also did not lead to a predictable next stimulus as with the *Scales* task. Indeed, for the *Free-to-Play* task, stimuli could appear at random times and in random order. Furthermore, the stimuli are only visual (the ball turning green and then red while resting on a key). Therefore, the user must continuously scan the playing area for visual cues, and the incentive to do so depends entirely on the user's ability. This can explain why the centre of the playing area is explored the most while the peripheral area is the least. Therefore, it can also explain why the keys in the centre are left unplayed for the shortest time on average. Moreover, also for this task, a noteworthy evolution is visible in the pattern of the average head direction of the different sessions. The patients appear to be exploring more as sessions in VR progress, as is the case for the *Memory* task. This can also indicate that some behaviour is taught to explore the neglected side. This behaviour provides additional evidence that the presented VR system contributes to the rehabilitation of patients with USN. Moreover, for this task, minimal mistakes were made by the test users, meaning that they rarely let a ball touch a key, and therefore, not a lot of data are available. This makes a comparison infeasible. This task aims explicitly to encourage patients to explore their neglected peripheral area. This was the reason for placing most keys on the neglected side instead of distributing them evenly as for the *Assessment* task. Comparable data from the *Assessment* would provide more insights into the behaviour of the patients and the impact of the VR tasks. Unfortunately, these data are not available from this study.

*8.2. Research Questions*

With the available information and insights, answers can be formulated to the research questions first presented in Section 3.

- **"Can MNT be effectively translated in a VR environment?"** The tests show promising results to support the application of MNT in VR. As indicated by the medical experts in the interviews, the combination of visual and auditory stimuli is beneficial. Combining both triggers and training the patients on both modalities teaches them connections between the two. The results also show some evidence that the addition of auditory stimuli impacts the performance of the exercises. On the one hand, in the *Memory* task, where users also respond to sound cues, half of the patients show some difference in performance based on where the stimuli are shown. That is, they perform worse for stimuli presented in the far most neglected side. On the other hand, in the *Free-to-Play* task, where users only respond to visual stimuli, 3 out of 4 patients show significantly worse performance, which can be linked to the USN. However, precaution is warranted with respect to the adoption of the proposed VR system for individuals susceptible to motion sickness and epilepsy in VR.

- **"Can the newly designed VR system be used for the treatment of adults with USN with an acquired brain injury?"** For the *Scales* task, not enough evidence is obtained that the mistakes made by the patient are the result of their USN. However, this does not mean that the task serves no purpose or does not train the patients in any way. The patients' attention is gradually guided towards the peripheral area on the neglected side. The data show that all patients successfully form this motion, and by doing so, they are urged to observe their neglected side as well. The patients performed quite differently in the *Memory* and *Free-to-Play* tasks. The results of these tasks do show the influence of the USN. In general, they performed worse on their neglected side. This provides sufficient evidence for the assumption that the patients were trained to perform better despite their USN, thereby treating the disorder. Furthermore, two of the patients were reported to perceive improvements in their performance of the different tasks. In conclusion, all patients could perform the exercises and no problems arose in the pilot study that would prevent this solution from being researched further. There is evidence to support the notion that the rehabilitation tasks have some effect

on the USN disorder in the patients. However, additional studies are needed to investigate the significance and persistence of these effects.

- **"Will the spatial performance of the patient with an acquired brain injury evolve positively after treatment with the VR system?"** The two-week limitation of this study prohibits a conclusive answer to this question. However, there is some evidence that suggests intra-session improvements in the exploration of the participants for the *Memory* and *Free-to-Play* tasks. Unfortunately, no insights are gathered on the persistence of this effect towards everyday life situations in the long term. One patient did report that they experienced improvements in daily life during the two-week period. It is also crucial to discuss what else could influence the apparent evolution of the head direction during sessions. Indeed, the gradual increase in the average head direction may result from taught behaviour during previous VR exercises. However, it is also possible that the change in the average head direction is the direct consequence of the different sessions themselves. Specifically, as the sessions progressed, the task's difficulty was often increased. A greater difficulty always meant that more notes were presented, and therefore a larger area of the playing area was needed. Thus, stimuli were presented in a larger area, which inherently would force users to explore more towards the neglected side as well. This would also result in an increase in the average head direction. Finally, it is also possible that the evolution of the behaviour in between sessions is not a result of the VR rehabilitation itself but of the rehabilitation the patients received in between VR sessions. Of course, a combination of any explanations is still possible.

*8.3. Limitations*

To perform a good analysis of the result in Section 7, it is important to understand the limitations and shortcomings of the study. Firstly, the sample size of $n = 8$ participants, of which four were clinically diagnosed USN patients, is relatively small. The statistical relevance of the results and the conclusions arising from them should be interpreted with a critical mind. Additional tests with clinical patients and non-clinical test users are required to further support the findings presented in this section. Secondly, the data collected during the tests solely focus on measuring the head direction and the task performance. Thirdly, the VR environment and the xylophone are currently abstract visualisations and can be extended in the future to create a more realistic experience. Furthermore, general prior knowledge of the participants is limited to the severity of USN. Therefore, recognising whether the recorded behaviour is induced by the USN or another underlying condition is impossible, e.g., the diagnosis of apraxia in Patient 4 could also impact the collected data. In this discussion, the assumption is adopted that any behaviour found in a patient is mainly the result of their USN diagnosis.

**9. Conclusions**

Currently used rehabilitation methods for USN often fail to take into account the heterogeneity of the syndrome by solely focusing on the visual modality. MNT shows great potential to bridge this gap, incorporating multiple modalities in the training process. The use of VR in rehabilitation has shown many advantages over traditional methods as it facilitates the modification of the therapy and allows for increased data collection regarding therapy progress.

Therefore, the authors designed, developed and evaluated a novel VR therapy tool for MNT. This paper investigated the feasibility of the tool for the rehabilitation of patients diagnosed with USN. The resulting proof of concept was evaluated during a two-week pilot study with four clinical patients and compared to the results of four non-clinical test users.

The results of the study indicated that the patients with USN were capable of performing the four rehabilitation tasks and indicated increased enjoyment. Next, the results showed that non-clinical test users performed significantly better than clinical users with

respect to task performance, which indicates that the system is able to discern between users with USN and users without USN. Furthermore, this result indicates that the tasks were challenging the patients. Nonetheless, for each task, the patients were able to respond to their entire environment and the presented triggers successfully. Over the course of 2 weeks, patients were shown to increasingly adjust their head direction towards their neglected side for the *Memory* and *Free-to-Play* tasks, presumably to improve their task performance. These results indicate that the VR tool managed to successfully translate MNT to a virtual environment and that the tool shows promise for the rehabilitation of USN. Moreover, the participating medical experts of this study responded positively to the developed tool. They believe that it will be a useful tool at an early rehabilitation stage. However, it should be noted that the sample size was limited to four clinical patients and four non-clinical test users. Additionally, the measures used in this study were solely focused on head rotation and task performance.

In conclusion, the tool developed by the authors is a valuable contribution to the current research on rehabilitation for USN as it is the first system to apply MNT in a VR environment. The promising results show increased enjoyment in the patients and sufficient challenges in the tasks. However, the tool and the evaluation have their limitations, specifically the limited sample size and limited metrics. With the enthusiasm to continue the research, future work should, therefore, focus on validating these results over a longer rehabilitation period with a larger sample. Moreover, due to the heterogeneity of the syndrome, future work should explore the incorporation of personalisation and gamification to fulfil the needs of the patients and further increase enjoyment during the rehabilitation process.

**Author Contributions:** Conceptualisation, J.H., S.C., E.V., C.V.L. and F.D.B.; methodology, J.H., S.C., E.V. and C.V.L.; software, E.V.; validation, J.H., S.C. and C.V.L.; formal analysis, J.H. and S.C.; investigation, S.C. and E.V.; data curation, J.H. and E.V.; writing—original draft preparation, J.H., S.C. and E.V.; writing—review and editing, J.H., S.C., C.V.L., F.D.B. and F.D.T.; visualisation, J.H. and S.C.; supervision, F.D.T. All authors have read and agreed to the published version of the manuscript.

**Funding:** This research received no external funding.

**Institutional Review Board Statement:** The study was conducted according to the guidelines of the Declaration of Helsinki and approved by the Ethics Committee of Ghent University Hospital on 19 March 2021 [B6702021000166].

**Informed Consent Statement:** Informed consent was obtained from all subjects involved in the study.

**Data Availability Statement:** The data are not publicly available due to privacy and ethical concerns.

**Acknowledgments:** We would like to express our gratitude to the Rehabilitation Centre of the Ghent University Hospital, the participants of this study and their surrounding team of therapists and specialists for helping us to accomplish this research.

**Conflicts of Interest:** The authors declare no conflict of interest.

## Abbreviations

| | |
|---|---|
| AR | Augmented Reality |
| BIT | Behavioural Inattention Test |
| CBS | Catherine Bergego Scale |
| CVS | Caloric Vestibular Stimulation |
| GVS | Galvanic Vestibular Stimulation |
| HMD | Head Mounted Display |
| LA | Limb Activation |
| MNT | Music Neglect Therapy |
| NMT | Neurologic Music Therapy |
| NVM | Neck-Muscle Vibration |

| OKS | Optokinetic Stimulation |
|-----|-------------------------|
| PA | Prism Adaptation |
| SAT | Sustained Attention Training |
| TAP | Test for Attention Performance |
| tDCS | Transcranial Direct Current Stimulation |
| TMS | Transcranial Magnetic Stimulation |
| USN | Unilateral Spatial Neglect |
| VR | Virtual Reality |
| VST | Visual Scanning Training |
| WHO | World Health Organisation |

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
