# Peer review of "From Patient to Musician: A Multi-Sensory Virtual Reality Rehabilitation Tool for Spatial Neglect"

_applsci, doi:10.3390/app12031242_

Round 1
Reviewer 1 Report
This paper discusses the musical neglect therapy with rehabilitation patients to make music. Immersive VR is applied for the rehabilitation purpose in the paper. A small test is conducted with over two weeks with 4 clinical users. Experiment results are positive and promising.
- The sample size is too small and thus the result is not really convincing. Larger experiment should be conducted after this preliminary study.
- Immersive VR element is fairly weak. Good example can be found in
Design and Development of a Virtual Dolphinarium for Children With Autism, IEEE Transactions on Neural Systems and Rehabilitation Engineering,
DOI:10.1109/TNSRE.2013.2240700. - Abstract and conclusion sections should be enhanced with contribution and limitation of the work better highlighted.
Reviewer 2 Report
Authors reported the use of a multi-sensory VR rehabilitation tool for spatial neglect. This topic is very interesting, look at these points:
- Lines 45-50: I'm not sure this part can be useful. I suggest to report at these point what is the aim of this paper.
- Lines 58-60: Actually, USN can also occur after a traumatic brain injury or after a total resetion of a brain tumor. Ref. --- Surgical outcome and molecular pattern characterization of recurrent glioblastoma multiforme: A single-center retrospective series. Clin Neurol Neurosurg. 2021 Aug;207:106735. doi: 10.1016/j.clineuro.2021.106735. --- Combining MGMT promoter pyrosequencing and protein expression to optimize prognosis stratification in glioblastoma. Cancer Sci. 2021 Sep;112(9):3699-3710. doi: 10.1111/cas.15024.
- The whole paragraph n ° 1 is very long and should be summarized a little bit. Please improve.
- Lines 308-400: "4. Design of the VR therapy for USN". Is this part the Materials and Method of this paper? If yes, please highlight this point. If no, is this part strictly necessary?
- Lines 483-486. Over what period of time were these patients selected?
- Lines 639-340: Limitations of the paper should be moved at the end of discussion
- Lines 220-223: "In Virtual Reality... using devices such as a Head Mounted Display (HMD) ..." The use of virtual reality and augmented reality in neurosurgery has increased exponentially in recent years. Please improve this point, look at these references. -- Evaluation of a Wearable AR Platform for Guiding Complex Craniotomies in Neurosurgery. Ann Biomed Eng. 2021 Sep;49(9):2590-2605. doi: 10.1007/s10439-021-02834-8. --- IBIS: an OR ready open-source platform for image-guided neurosurgery. Int J Comput Assist Radiol Surg. 2017 Mar;12(3):363-378. doi: 10.1007/s11548-016-1478-0.
- Line 627-635: "The experts believed that the proposed therapy tool could treat neglect in.. and immersion of the therapy were considered beneficial. " Which experts are the authors referring to?
- Lines 694-695: "Bernardi et al. [15] which stated that individuals performed better under conditions with congruent sound feedback." Enhance this point.
Reviewer 3 Report
The manuscript addresses a timely and relevant study and is already in a mature state. I would just like to make some proposals which could help to improve the manuscript within in a revision round (my recommendation: minor revisions).
- In your section 2.5 (“VR for rehabilitation”, third paragraph), you refer to investigated effects which have an impact on the usage on VR and performance within it. I agree that motion sickness effects are still a problematic issue leading to a broad problem. However, in research on spatial cognition, there are also recent studies available which point to distorting effects in terms of spatial perception and memory, such as wrongly estimated distances, depending on the locomotion technique used (https://doi.org/10.3390/ijgi10030150 and ). This indicates potentials to further explore motion within in immersive virtual environments, with potential effects on spatial measures in therapy applications. Being able to estimate distances accurately is highly relevant for an accurate formation of cognitive representations of (geographic) space (see https://doi.org/10.1038/nn.4656).
- In your methodology, you write about a changing color when the keys were hit. How long did they last in this changed color, and don’t you see an influencing effect when the color arrangement and design of the keys are changed over a longer time?
- What I miss in your (quite long and well-structured) discussion section is the transfer of your study results to the state-of-the-art literature. In how far do your study results back up, extend or even contradict previously published studies? While your introduction and background is based on a stable body of literature, the important placement of your study results within the ongoing debates are missing so far.
Round 2
Reviewer 1 Report
The revised paper has several improvements. Yet the abstract and conclusion sections can be enhanced to highlight the contribution, limitation and future efforts.
Reviewer 2 Report
Authors solved all my criticisms.
Author Response
Dear reviewer,
Thank your for your feedback. We are glad all your concerns were resolved by our revised version.
Kind regards,
The authors